# Optical Temperature-Sensing Performance of La_2_Ce_2_O_7_:Ho^3+^ Yb^3+^ Powders

**DOI:** 10.3390/ma17071692

**Published:** 2024-04-07

**Authors:** Jiameng Chao, Hui Lin, Dechao Yu, Ruijin Hong, Zhaoxia Han, Chunxian Tao, Dawei Zhang

**Affiliations:** Engineering Research Center of Optical Instrument and System, Ministry of Education and Shanghai Key Laboratory of Modern Optical System, University of Shanghai for Science and Technology, No. 516 Jungong Road, Shanghai 200093, China; jiameng_chao@163.com (J.C.); hanzhaoxia0810@163.com (Z.H.); dwzhang@usst.edu.cn (D.Z.)

**Keywords:** optical temperature sensing, up-conversion emission, energy transfer, pyrochlore structure

## Abstract

In this paper, La_2_Ce_2_O_7_ powders co-activated by Ho^3+^ and Yb^3+^ were synthesized by a high temperature solid-state reaction. Both Ho^3+^ and Yb^3+^ substitute the La^3+^ sites in the La_2_Ce_2_O_7_ lattice, where the Ho^3+^ concentration is 0.5 at.% and the Yb^3+^ concentration varies in the range of 10~18% at.%. Pumped by a 980 nm laser, the up-conversion (UC) green emission peak at 547 nm and the red emission at 661 nm were detected. When the doping concentration of Ho^3+^ and Yb^3+^ are 0.5 at.% and 14% at.%, respectively, the UC emission reaches the strongest intensity. The temperature-sensing performance of La_2_Ce_2_O_7_:Ho^3+^ with Yb^3+^ was studied in the temperature range of 303–483 K, where the highest relative sensitivity (S_r_) is 0.0129 K^−1^ at 483 K. The results show that the powder La_2_Ce_2_O_7_:Ho^3+^, Yb^3+^ can be a potential candidate for remote temperature sensors.

## 1. Introduction

Since the concept of UC luminescence was first proposed by the French scientist Auzel in 1966 [1], it has been widely concerned in the application fields of remote thermometers [2], stimulated emission depletion [3], solar cells [4], etc. Remote thermometers based on UC luminescent materials are suitable for harsh environments, such as strong magnetic fields and corrosion [5]. Theoretically, the fluorescence intensity, fluorescence intensity ratio (*FIR*), fluorescence lifetime, and full width at half maximum (FWHM) of luminescent materials will change with the variation in temperature [6]. Based on these characteristics, the purpose of temperature sensing can be feasibly realized.

The optical properties of fluorescent materials are usually temperature-dependent within a certain temperature range. However, it should be pointed out that the measurement of fluorescence lifetime and intensity is highly dependent on the intensity of excitation light source, background noise, and other factors, so it is difficult to ensure the accuracy of the temperature measurement based on the above factors. At present, most of the research works on optical temperature measurement are based on *FIR* with good anti-interference ability [7,8,9,10,11]. Moreover, the UC luminescent materials activated by rare earth ions usually have abundant but narrow linear luminescence bands, which naturally feature more excellent *FIR*, and make it possess unique advantages in the field of temperature sensing.

UC luminescent materials for optical temperature measurement need to have high luminescence intensity, for which the selection of the host materials is very important. In recent years, the research on A_2_B_2_O_7_ compounds has been blooming, a general formula that describes most pyrochlores as A_2_B_2_O_6_O’, where the A site is occupied by a 3+ or 2+ cation with eight-fold coordination, the B site is occupied by a 4+ or 5+ cation with six-fold coordination, and there are two distinct oxygen atom sites [12]. At present, the reported A_2_B_2_O_7_ compounds are mainly La_2_Ce_2_O_7_ [13], Gd_2_Ce_2_O_7_ [14], Nd_2_Ce_2_O_7_ [15], etc., and the main application scenario focuses on the application of a thermal barrier coating, but few reports are found in the field of temperature sensing based on the fluorescence performance. La_2_Ce_2_O_7_ belongs to the structure of defective fluorite, where the A site is occupied by La^3+^, and the B site is occupied by Ce^4+^, as shown in Figure 1a. La_2_Ce_2_O_7_ has a high melting point (>2000 °C), excellent chemical stability, and high temperature phase stability and catalytic performance, which is widely used in thermal barrier coatings, infrared radiation ceramic materials, catalytic materials, etc. [16,17]. However, reports on the optical temperature-sensing performance of rare earth-doped La_2_Ce_2_O_7_ materials are few [18,19].

Due to the small energy gap between ^2^H_11/2_ and ^4^S_3/2_ of the Er^3+^ ions, the *FIR* conforms to the Boltzmann distribution and has a high sensitivity to temperature variation; Er^3+^ has been most extensively investigated by researchers [19,20]. In addition, in recent years, there have been numerous studies on the temperature-sensing performance of Ho^3+^-activated phosphor materials [21,22]. Particularly, some studies have shown that Ho^3+^-doped UC luminescent materials have excellent temperature-sensing performance [23,24]. Ho^3+^ can efficiently emit colorful UC fluorescence with the assistance of a Yb^3+^ sensitizer under an excitation of the 980 nm near-infrared laser [25,26]. Govind B. Nair et al. [27] prepared Ho^3+^/Yb^3+^ co-doped BaY_2_F_8_, and investigated the UC luminescence with temperature change in the temperature range of 303–623 K, and obtained the maximum relative sensitivity (S_r_) of 0.006051 K^−1^ at 303 K.

In this work, a series of La_2_Ce_2_O_7_:Ho^3+^ Yb^3+^ powders were prepared by high temperature solid-state reaction. The temperature dependence of the UC emission intensity has been investigated. The results show that La_2_Ce_2_O_7_:Ho^3+^ Yb^3+^ powders are good candidates for optical temperature sensing.

## 2. Experimental

### 2.1. Materials Preparation

High purity powder Ho_2_O_3_ (99.99%), Yb_2_O_3_ (99.99%), CeO_2_ (99.99%), and La_2_O_3_ (99.999%) were used as raw materials and weighed according to the chemical composition of (Ho_0.005_Yb_x_La_0.995−x_)_2_Ce_2_O_7_ (x = 0.10, 0.12, 0.14, 0.16, 0.18). Since La_2_O_3_ is easy to absorb moisture, it is kept in the oven at 120 °C before use. The mixed powders were ground with an agate grinder for 1 h. The resultant powder mixtures were thermally treated in a Muffle furnace by the following sintering procedure: raised to 1370 K from room temperature within 150 min, then raised to 1890 K within 120 min and kept for 300 min, and finally cooled down to room temperature naturally. For the following characterizations, the obtained products were ground into fine powders with an agate grinder for 1 h.

### 2.2. Characterizations

X-ray diffraction (XRD, Rigaku SmartLab, Japan, Akishima City, Tokyo) was used to analyze the phase and crystal structure of the (Ho_0.005_Yb_x_La_0.995−x_)_2_Ce_2_O_7_ (x = 0.10, 0.12, 0.14, 0.16, and 0.18) samples. The scanning range is 20–90°, the step size is 0.02°, and the scanning speed is 8°/min. The UC emission spectra under excitation of a 980 nm laser were recorded by a fluorescence spectrometer (FLS1000, Edinburg Instruments, UK, Edinburgh) with a resolution of 0.05 nm. In addition, the temperature-dependent PL spectra were measured in the 303–483 K temperature range, and the thermal recovery property and repeatability of the samples were also measured on the fluorescence spectrometer.

## 3. Results and Discussion

Figure 1b shows the XRD θ-2θ scanning patterns of La_2_Ce_2_O_7_ and (Ho_0.005_Yb_x_La_0.995−x_)_2_Ce_2_O_7_ (x = 0.10, 0.12, 0.14, 0.16, and 0.18) powder samples thermally treated at 1890 K for 5 h. The diffraction patterns of all the Ho^3+^ and Yb^3+^ co-doped samples were the same as La_2_Ce_2_O_7_, without other phases detected. The difference is that the diffraction peaks are all shifted to higher angles, indicating that Ho and Yb have been successfully doped into the lattice of La_2_Ce_2_O_7_. The shift of the diffraction peaks to a higher angle is caused by lattice shrinkage by the substitution of La^3+^ (*R_La_^3+^* = 1.22 Å) ions with larger ionic radii by Ho^3+^ (*R_Ho_^3+^* = 1.03 Å) and Yb^3+^ (*R_Yb_^3+^* = 0.887 Å) ions with a smaller ionic radius.

Figure 2a shows the UC emission spectra of the (Ho_0.005_Yb_x_La_0.995−x_)_2_Ce_2_O_7_ (x = 0.10, 0.12, 0.14, 0.16, and 0.18) powders under the excitation of a 980 nm laser (pumping power 150 mW) at room temperature. Although the doping amount of Yb^3+^ is various, the UC emission is mainly composed of a strong green emission peak at 547 nm (Ho^3+^: ^5^S_2_/^5^F_4_→^5^I_8_) and a weak red emission peak at 661 nm (Ho^3+^: ^5^F_5_→^5^I_8_). When the substitution amount of Yb^3+^ reaches x = 0.14, both the UC intensity of the green and red reach the maximum value. And, when x > 0.14, the UC intensity was gradually decreased, which was caused by the energy loss in the energy transfer (ET) between the Yb^3+^ ions or during the Yb^3+^→Ho^3+^ ET process.

Figure 2b shows the UC emission intensity spectra of the (Ho_0.005_Yb_0.14_La_0.855_)_2_Ce_2_O_7_ sample under the excitation of a 980 nm laser with different pumping powers (200–2140 mW) at room temperature. It can be clearly seen that with the increase in laser power, the 547 nm UC emission intensity (I_547_) and the 661 nm UC emission intensity (I_661_) rapidly increases. The initial UC emission intensity increases with the increase in excitation power, then gradually decreases when the power reaches 1800 mW. When the excitation power is less than 1400 mW, the UC emission intensity changes rapidly, and when the power is more than 1400 mW, the UC emission intensity changes slowly. A hysteresis-like loop was observed for the Ho^3+^: ^5^F_4_/^5^S_2_→^5^I_8_ transitions of ions when the pump power was increased from 200 mW to 2140 mW for the (Ho_0.005_Yb_0.14_La_0.855_)_2_Ce_2_O_7_ sample, and, conversely, for the excitation optical power down process. It can be seen from Figure 2b that with the pumping power increased, the UC emission intensity first increases and then decreases owing to the multi-phonon assisted non-radiative relaxations. However, in the “power-down” process, it does not take the same path, but rather follows a path with a slightly lower intensity and gives rise to a hysteresis curve. The same output power corresponds to two different emission intensities, and there is a loss in the intensity in the reverse process. The obtained hysteresis loop is clarifying the evidence of intrinsic optical bi-stability (IOB). The existence of emission intensity lag within a complete excitation power period is an indicator of inherent optical bi-stability [28]. At the same time, the increase in temperature caused by the increase in laser power is also one of the reasons for the hysteresis cycle of UC emission intensity.

According to the UC emission intensity theory, the UC emission intensity is proportional to the pumping power intensity, where n is the number of photons required for UC emission. In the UC emission process, the relationship between UC emission intensity (*I*) and excitation power (*P*) [29,30,31] can be expressed by the following formula:(1)$$I\propto P^{n}$$

The logarithm of pump power and the logarithm of luminescence intensity are used for linear fitting, and the slope of the line obtained is the number of photons required for UC emission, *n*, which can be obtained by the logarithm of Equation (1):(2)$$n=\frac{LnI}{LnP}$$

For the (Ho_0.005_Yb_0.14_La_0.855_)_2_Ce_2_O_7_ sample, the logarithmic relationship plotting between the intensity of the 547 nm and 661 nm UC emissions and the excitation power *P* is shown in Figure 2c. The slopes of the linear fitting for the 547 nm and 661 nm UC emissions are 1.23 and 1.15, respectively, indicating that both of the two UC emissions are based on the two-photon process. For the two-photon process, *n* should be equal to or approximately equal to 2, but the n value of both sets of experimental data are less than 2. This may be caused by the competition between linear decay and UC processes for the depletion in the intermediate excited states [32]. As reported by Xue et al. [33], when UC is dominant, the intensity is proportional to the excitation power density, and when the intermediate excited state energy level is dominated by linear decay, the intensity is proportional to the square of the excited power. Therefore, since *n* is closer to 1 (*n* = 1.23, 1.15), the linear decay of the intermediate excited state is dominant in this work.

In order to better understand the UC emission process of La_2_Ce_2_O_7_:Ho^3+^ Yb^3+^, the energy level diagrams of Ho^3+^ and Yb^3+^ and the possible energy transfer during the UC emission process are provided in Figure 2d. Under the excitation of a 980 nm laser, the Yb^3+^:^2^F_7/2_ ground state electrons absorb the energy and transit to the Yb^3+^:^2^F_5/2_ excited state, while the Ho^3+^:^5^I_8_ ground state electrons jump to the Ho^3+^:^5^I_6_ excited state through the energy transfer (ET1) from Yb^3+^ to Ho^3+^. The intermediate level of Ho^3+^:^5^I_6_ has a long life, which can continue to absorb energy and, in the meantime, transit to the excited state of Ho^3+^:^5^F_4/5_/^5^S_2_ of higher energy via excited state absorption (ESA). Some electrons decay from the Ho^3+^:^5^I_6_ state to the Ho^3+^:^5^I_7_ state through multi-phonon relaxation. Electrons in the Ho^3+^:^5^I_7_ state can go up to the Ho^3+^:^5^F_5_ state, which is also an ESA process. Finally, electrons transit from the Ho^3+^:^5^F_4/5_/^5^S_2_ and Ho^3+^:^5^F_5_ states to the Ho^3+^:^5^I_8_ ground state. Above all, the green and red UC emissions are generated.

In order to study the temperature-sensing performance of Yb^3+^ and Ho^3+^ co-doped La_2_Ce_2_O_7_ UC-emitting powders, temperature-dependent UC emission spectra under the excitation of a 980 nm laser were measured in the range of 303–483 K, as shown in Figure 3a. The inset shows the enlarged view of the red UC emission spectra in the wavelength range of 640–680 nm. It can be observed that with the increase in temperature, the intensity of the green and the red UC emissions gradually weakens, due to the thermal quenching. With the increase in temperature, phonon vibration in the lattice is enhanced, leading to the increase in the probability of non-radiative relaxation, and the excessive energy is lost in the form of heat, resulting in the reduced UC emission intensity [10,34,35].

Specifically, as shown in Figure 3b, I_661_ has no obvious tendency to change with temperature; however, I_547_ decreased evidently when the temperature increased from 303 K to 483 K, which may be because the probability of non-radiative relaxation from ^5^F_4_/^5^S_2_ level to ^5^F_5_ level increased with increasing temperature. At the same time, the increase in temperature will also produce thermal excitation to promote the electron transition to the higher energy level of ^5^F_4_/^5^S_2_. The different response of these two energy levels to temperature provides favorable conditions for obtaining excellent temperature sensitivity [36].

Non-thermally coupled energy levels involve two independent excited energy levels, each with a unique temperature dependence. *FIR* based on non-thermally coupled levels of ^5^F_4_/^5^S_2_→^5^I_8_ and ^5^F_5_→^5^I_8_ can be used for optical temperature measurement. Since the intensity of 547 nm and 661 nm UC emissions are temperature-dependent, the *FIR* of corresponding energy levels can be used for temperature sensing. Figure 3c shows the plot of the *FIR* of the 661 nm and 547 nm UC emissions against the absolute temperature. The ratio I_661_/I_547_ increased from 0.13 to 0.46 with the temperature increasing from 303 K to 483 K. Figure 3d shows the plot of Ln (I_661_/I_547_) as a function of the inverse absolute temperature, which presents a straight line with a slope of −967.1158 and an intercept of 1.08179. In addition, the sensitivity is a very major parameter to estimate the performance of the sensor. The absolute sensitivity (*S_a_*) and the relative sensitivity (*S_r_*) can be given as the following [37]:(3)$$S_{a}=\left|\frac{dFIR}{dT}\right|$$
(4)$$S_{r}=\left|\frac{1}{FIR}\frac{dFIR}{dT}\right|$$

Figure 4a,b show the variation in the *S_a_* and *S_r_* of samples with temperature within the temperature range of 303–483 K. It can be seen that the absolute sensitivity *S_a_* of the sample increases with the increase in temperature. Under the excitation of a 980 nm laser, the maximal *S_a_* value is 0.0059 K^−1^ at 483 K. *S_r_* shows fluctuating variation with the temperature, reaching a maximal value of 0.0129 K^−1^ around 483 K.

Temperature resolution (*δT*) is an important parameter for evaluating the performance of temperature sensing, indicating the smallest detectable temperature change. The temperature resolution obtained through experiments is generally expressed as [38]
(5)$$\delta T=\frac{1}{S_{r}}\frac{\delta FIR}{FIR}$$

In the above equation, *δFIR* represents the precision of the *FIR*, while *δFIR*/*FIR* is primarily determined by the properties of the detector. By conducting three measurements of the emitted spectrum and calculating their variances, the standard deviation of *FIR* at room temperature is determined to be 0.0378%. Utilizing this value, the temperature resolution can be computed, as illustrated in Figure 4c. It can be observed that the *δT* values range between 0.064 and 0.52, with an average *δT* value of around 0.25 K.

In order to investigate the thermal recovery performance of (Ho_0.005_Yb_0.14_La_0.855_)_2_Ce_2_O_7_ powder, five emission spectra were measured under 1400 mW: (a) when the temperature rises to 413 K, the emission spectrum was measured immediately; (b) when 413 K lasts for 5 min; (c) when 413 K lasts for 10 min; (d) when 413 K lasts for 20 min; and (e) when 413 K lasts for 40 min. As shown in Figure 4d, after 413 K lasts for 5 min, the UC emission intensity decreased significantly, but after holding the powder for 10 min, 20 min, and 40 min, the UC emission intensity was gradually increased and recovered to the height of “(a)”, indicating that the thermal recovery performance of La_2_Ce_2_O_7_ powder is good.

The optical temperature-sensing performance of materials doped with Ho^3+^ and Yb^3+^ based on the *FIR* technology are shown in Table 1. The relative sensitivity of (Ho_0.005_Yb_0.14_La_0.855_)_2_Ce_2_O_7_ phosphor is higher than that of most of the materials listed in Table 1, which shows that Ho^3+^ and Yb^3+^-doped La_2_Ce_2_O_7_ powder is a potential candidate for optical temperature sensing.

## 4. Conclusions

In this work, Ho^3+^ and Yb^3+^ co-doped La_2_Ce_2_O_7_ powders were prepared by solid-state reaction at 1890 K. Under excitation by a 980 nm laser, green UC emission peaks were observed at 547 nm and red UC emission peaks at 661 nm at room temperature, with the UC emission intensity decreasing as the temperature increased. A quite clear relationship can be observed between the Ho^3+^/Yb^3+^ UC emission and the temperature. The highest relative sensitivity was 0.0129 K^−1^ at the temperature of 483 K. These properties indicate that La_2_Ce_2_O_7_:Ho^3+^ Yb^3+^ powder is a potential candidate material for optical temperature sensing.

## Figures and Tables

**Figure 1 materials-17-01692-f001:**
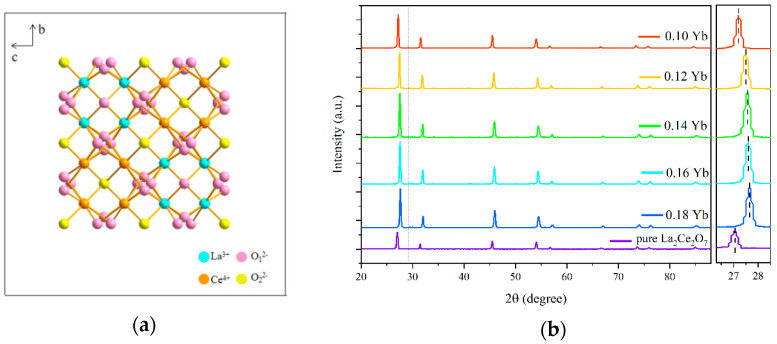
(**a**) A schematic diagram of La_2_Ce_2_O_7_ crystal structure. (**b**) XRD θ-2θ scanning patterns of the La_2_Ce_2_O_7_:Ho^3+^, Yb^3+^ powders with different doping concentrations of Yb.

**Figure 2 materials-17-01692-f002:**
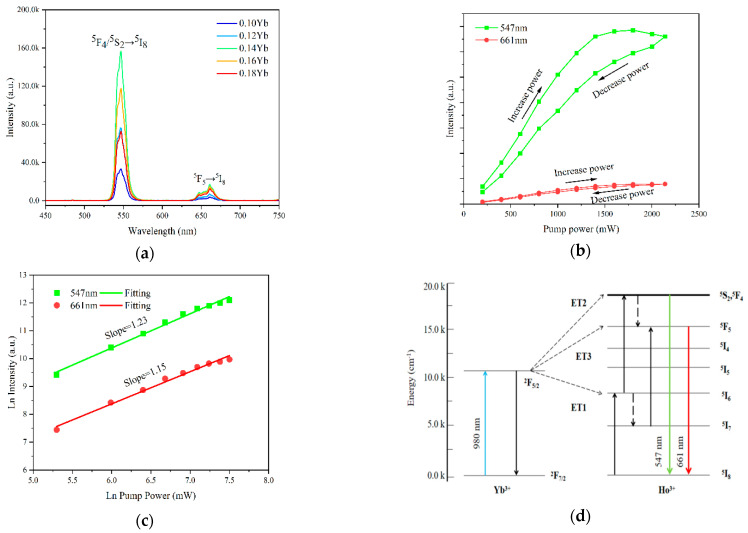
(**a**) UC emission spectra of the (Ho_0.005_Yb_x_La_0.995−x_)_2_Ce_2_O_7_ (x = 0.10, 0.12, 0.14, 0.16, and 0.18) powders under the excitation of a 980 nm laser. (**b**) The UC emission intensity of the (Ho_0.005_Yb_0.14_La_0.855_)_2_Ce_2_O_7_ in a power cycle (increase power and decrease power) varies with different pumping power (200 mW–2140 mW) laser excitation. (**c**) The pumping power dependence of the UC emission intensity of the (Ho_0.005_Yb_0.14_La_0.855_)_2_Ce_2_O_7_ sample under the excitation of a 980 nm laser. (**d**) Schematic energy level diagram of Yb^3+^ and Ho^3+^ ions and proposed energy transfer routes for the UC emission.

**Figure 3 materials-17-01692-f003:**
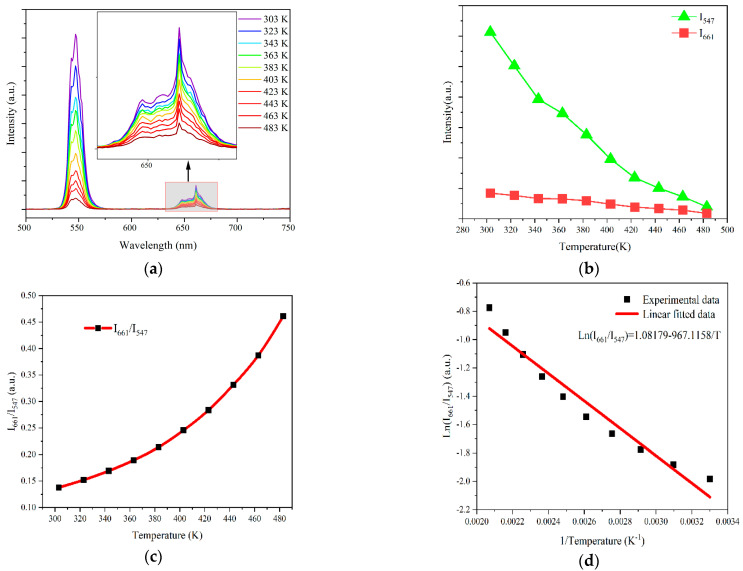
(**a**) Temperature-dependent UC emission spectra of the (Ho_0.005_Yb_0.14_La_0.855_)_2_Ce_2_O_7_ powder under the excitation of a 980 nm laser. (**b**) The green and red UC emission intensity as a function of temperature. (**c**) Temperature dependence of the UC emission spectra intensity ratio (I_661_/I_547_) of 661 nm and 547 nm emissions. (**d**) Ln (I_661_/I_547_) as a function of the inverse absolute temperature.

**Figure 4 materials-17-01692-f004:**
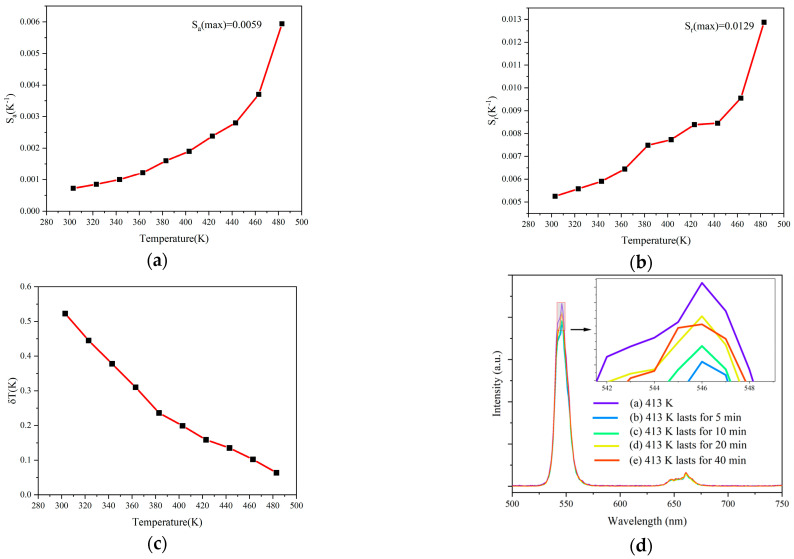
(**a**) Absolute sensitivity of the (Ho_0.005_Yb_0.14_La_0.855_)_2_Ce_2_O_7_ powder at various temperatures. (**b**) Relative sensitivity of the (Ho_0.005_Yb_0.14_La_0.855_)_2_Ce_2_O_7_ powder at various temperatures. (**c**) Temperature resolution (*δT*). (**d**) Thermal recovery property of the (Ho_0.005_Yb_0.14_La_0.855_)_2_Ce_2_O_7_ powder.

**Table 1 materials-17-01692-t001:** Optical temperature-sensing properties of Ho^3+^/Yb^3+^-doped materials.

Materials	Temperature Range (K)	λ_ex_ (nm)	*S_r_* (K^−1^) (Max)	References
NaLaMgWO_6_	293–553	980	0.01079 (508 K)	[2]
Ba_0.77_Ca_0.23_TiO_3_	93–300	980	0.0053 (93 K)	[10]
Tellurite glass	303–503	975	0.011 (503 K)	[24]
BaTiO_3_	303–513	980	0.0034 (303 K)	[25]
CaMoO_4_	303–543	980	0.0066 (353 K)	[26]
LiYF_4_	100–500	976	0.0129 (150 K)	[39]
Gd_0.74_Y_0.2_TaO_4_	330–660	980	0.0037 (660 K)	[40]
LaOF	298–548	980	0.00451 (298 K)	[41]
Y_2_O_3_	303–623	980	0.0064 (423 K)	[42]
(Ho_0.005_Yb_0.14_La_0.855_)_2_Ce_2_O_7_	293–428	980	0.0129 (428 K)	This work

## Data Availability

The data of this paper are available on request from the corresponding author.

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
