# Peer review of "Optical Temperature-Sensing Performance of La2Ce2O7:Ho3+ Yb3+ Powders"

_materials, 2024, doi:10.3390/ma17071692_

Round 1

Reviewer 1 Report

Comments and Suggestions for Authors

In this work, the up-conversion luminescence processes in La2Ce2O7 powders co-activated by Ho3+ and Yb3+ ions and their mechanisms have been studied. In particular, Up-conversion luminescence has been examined for temperature sensor applications. In my opinion, the results are interesting and should be published in the Materials after minor revision.

1. Part 2.2: Resolution for emission spectra measurements should be given.

2. The following sentences in the text (lines 149-152): "(c) Pumping power 149 dependence of the UC emission intensity of the (Ho0.005Yb0.14La0.855)2Ce2O7 sample under 150 excitation of a 980 nm laser. (d) Schematic energy level diagram of Yb3+ and Ho3+ ions and 151 proposed energy transfer routes for the UC emission." should be deleted (this is figure caption).

3. Next sentence (lines 153-155) should be improved:

For the (Ho0.005Yb0.14La0.855)2Ce2O7 sample, the logarithmic relationship plottings between the intensity of the 547 nm and 661 nm UC emissions and the excitation power P is 154 shown in Figure 2(c) (not Figure 2(d)).

4. For better clarity and potential readers, the quality of figures, especially Fig. 3(a), Fig. 4(c), should be improved. The insets of the figures are nearly invisible.

5. Table 1 (the last column references) should be improved.

Reviewer 2 Report

Comments and Suggestions for Authors

Reviewer 3 Report

Comments and Suggestions for Authors

Manuscript ID: materials-2944604

Title: Optical Temperature Sensing Performance of La2Ce2O7:Ho3+, Yb3+ powders

Authors: Jiameng Chao, Hui Lin, Dechao Yu, Ruijin Hong, Zhaoxia Han, Chunxian Tao, Dawei Zhang

The topic of the manuscript may be interesting, but it needs some major revisions. To become publishable, at least the following aspects must be considered:

1. In abstract, lines 11-14, the dopant ion concentrations are not written properly. For example, instead of “0.05% at.%” it should be written “0.05 at.%”.

2. The authors selected/fixed the concentration of Ho3+ ions at 5 at.%. A comment explaining why this concentration was selected/fixed should be added to the manuscript.

3. The authors state that the obtained compounds were investigated using powder-form samples. What powder support device was used in the spectroscopic measurements?

4. The text in the lines 159-165 is not at all clear. It should be rewritten.

5. The text in lines 215-217 is difficult to understand. I recommend the following writing: “..., five emission spectra were measured for a pump power of 1400 mW: (a) when the temperature rises to 413 K, the emission spectrum was measured immediately, (b) 413 K lasts for 5 min, (c) 413 K lasts for 10 min, (d) 413 K lasts for 20 min, and (e) 413 K lasts for 40 min.”

6. The figure caption of Fig. 2 (lines 105-111) is found for the second time in the manuscript (lines 146-152) and should be removed.

7. The sentence in lines 225-226 (“... of Ho3+ and Yb3+ ions doped with other emitting materials activated by Ho3+ and Yb3+ ...” should be rewritten.

8. There are some mistakes in the manuscript as follows:

- lines 42-43 instead of +3, +2, +4, and +5 it should be written 3+, 2+, 4+, and 5+, respectively;

- line 61: “[24,25] Particularly”;

- line 69: “solid phase”;

- line 121: “radually”;

- line 126: “transitions of ion”;

- line 127: “sample,and”;

- line 130: “relaxations.However”;

- line 191: “. Specifically”;

- line 234: “with the under”.

Comments on the Quality of English Language

The English Language could be improved.
